# Extremely high magnetoresistance and conductivity in the type-II Weyl semimetals $WP_2$ and $MoP_2$

Nitesh Kumar[1], Yan Sun[1], Nan Xu [2], Kaustuv Manna[1], Mengyu Yao[2], Vicky Süss[1], Inge Leermakers [3], Olga Young[3], Tobias Förster[4], Marcus Schmidt[1], Horst Borrmann[1], Binghai Yan[5], Uli Zeitler[3], Ming Shi[2], Claudia Felser[1] & Chandra Shekhar[1]

The peculiar band structure of semimetals exhibiting Dirac and Weyl crossings can lead to spectacular electronic properties such as large mobilities accompanied by extremely high magnetoresistance. In particular, two closely neighboring Weyl points of the same chirality are protected from annihilation by structural distortions or defects, thereby significantly reducing the scattering probability between them. Here we present the electronic properties of the transition metal diphosphides, $WP_2$ and $MoP_2$, which are type-II Weyl semimetals with robust Weyl points by transport, angle resolved photoemission spectroscopy and first principles calculations. Our single crystals of $WP_2$ display an extremely low residual low-temperature resistivity of $3\,n\Omega\,cm$ accompanied by an enormous and highly anisotropic magnetoresistance above 200 million % at 63 T and 2.5 K. We observe a large suppression of charge carrier backscattering in $WP_2$ from transport measurements. These properties are likely a consequence of the novel Weyl fermions expressed in this compound.

[1] Max Planck Institute for Chemical Physics of Solids, 01187 Dresden, Germany. [2] Paul Scherrer Institute, Swiss Light Source, CH-5232 Villigen PSI, Switzerland. [3] High Field Magnet Laboratory (HFML-EMFL), Radboud University, Toernooiveld 7, 6525 ED Nijmegen, The Netherlands. [4] Dresden High Magnetic Field Laboratory (HLD-EMFL), Helmholtz-Zentrum Dresden-Rossendorf, 01328 Dresden, Germany. [5] Department of Condensed Matter Physics, Weizmann Institute of Science, Rehovot 7610001, Israel. Correspondence and requests for materials should be addressed to N.K. (email: Nitesh.Kumar@cpfs.mpg.de) or to C.F. (email: Claudia.Felser@cpfs.mpg.de)

Recently, many semimetals were found to exhibit Weyl and Dirac crossings in their band structure and, as a consequence high magnetoresistance and high mobilities[1–3]. Dirac points are four-fold degenerate[4–6] whereas Weyl points are two-fold degenerate and come in pairs with opposite chirality, namely, a source and a sink of the Berry curvature[7,8]. In 2012, Na₃Bi was the first semimetal predicted to contain Dirac fermions which was soon experimentally verified[5,9]. The first Weyl semimetal was anticipated[8] in 2015 and then quickly discovered in TaAs[10], and its close relatives[11,12]. Later, type-II Dirac and Weyl fermions (WSM-II) were identified, in which the Dirac or Weyl cones, respectively, are tilted with respect to the Fermi energy[13–15]. The archetypical WSM-IIs are the two dimensional van der Waals compounds WTe₂ and MoTe₂[13,16–19], in which pairs of neighboring Weyl points have opposite chirality. Fermions with even higher degeneracy can be found in compounds with certain symmetries where atoms sit on Wyckoff positions with high multiplicities[20]. For a compound to display Weyl points it must exhibit either inversion symmetry breaking, as in, for example, TaAs[8], or time reversal symmetry breaking, as in, for example, GdPtBi[21,22]. In these compounds, the Weyl points (WPs) of opposite chirality are close to each other and, hence, are vulnerable to annihilation from structural distortions or defects.

Very recently, the three dimensional transition metal diphosphides, WP₂ and MoP₂, were predicted to host four pairs of type-II WPs below the Fermi energy[23]. One important characteristic feature of these WSM-IIs are that the nearest WPs are of the same chirality and, therefore, are robust against structural distortions or defects. One might then expect high conductivities which indeed we have discovered in WP₂. Here we show that WP₂ exhibits extremely high conductivity and the highest magnetoresistance (*MR*) values yet observed in any compound with an extraordinarily large mean free path of 0.5 mm. A similar effect is also observed in MoP₂.

## Results

**Structure of tungsten and molybdenum diphosphides.** WP₂ is a three dimensional compound which crystallizes in a non-symmorphic $Cmc2_1$ space group. In this orthorhombic structure, tungsten atoms are surrounded by seven P atoms, six located at the corners of triangular prism and the seventh outside one of the rectangular faces. As can be seen in Fig. 1a the compound contains a mirror plane perpendicular to the *a*-axis, a *c*-glide perpendicular to the *b*-axis and a two-fold screw axis along the *c*-axis. Interestingly, the space group symmetry of WP₂ is very similar to the two dimensional WTe₂ which also contains a mirror-plane, a glide-plane and a two-fold screw axis[13]. MoP₂ crystallizes in the same structure. Figure 1b shows a typical crystal of WP₂ which is needle-shaped with its length oriented along the *a*-axis.

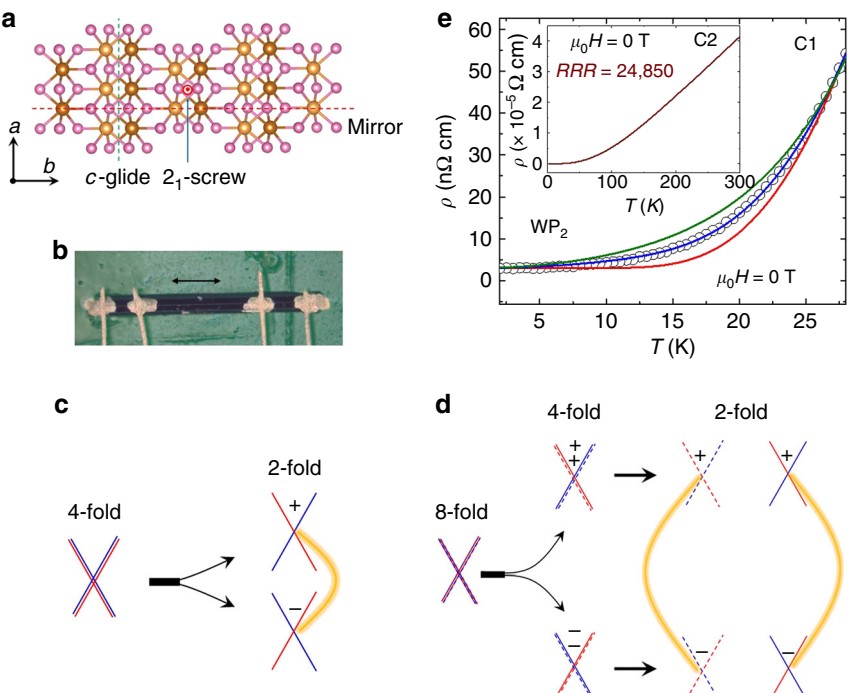

**Fig. 1** Crystal structure, *T*-dependent resistivity and evolution of Weyl points in W/MoP₂. **a** Non-symmorphic crystal structure of W/MoP₂ (W/Mo and P atoms are denoted by brown and pink spheres, respectively). For clarity, W/Mo atoms on the upper *ab*-plane are denoted by light brown spheres and the W/Mo atoms on the *ab*-plane displaced by half of the unit cell along the c-axis are denoted by dark brown spheres. Red and green dashed lines are the positions of the mirror and glide planes, respectively. Screw-axis along the c-axis is shown by red dot. **b** A needle shaped single crystal of WP₂ with the length, depth and width along *a*, *b* and *c*-axis, respectively. The scale bar is equivalent to 300 μm. **c** Splitting of a four-fold degenerate Dirac point with zero Chern number into 2, two-fold degenerate Weyl points of opposite chirality upon symmetry breaking. **d** The neighboring Weyl points in W/MoP₂ have same chirality. This kind of Weyl point can be viewed as the splitting of an 8-fold degenerate point without inclusion of SOC. On reducing the symmetry, the 8-fold linear crossing split into a pair of 4-fold degenerate points with opposite Chern numbers of C = ± 2, which is just the overlap of two Weyl point with same chirality. SOC just lifts the overlap of two Weyl points in *k*-space. Since the Weyl points with opposite chirality are relatively far away from each other, they are more robust and the Fermi arcs are longer than those formed in normal Weyl semimetals. **e** Low temperature resistivity of WP₂ at zero magnetic field. Green solid line is a fit of the $\rho(T)$ data with electron-electron scattering and electron–phonon scattering terms ($\rho(T) = \rho_0 + a^*T^2 + b^*T^5$); red solid line is a fit of the $\rho(T)$ data with phonon drag term ($\rho(T) = \rho_0 + c^*exp(-T_0/T)$); blue solid line is a fit considering all the above terms ($\rho(T) = \rho_0 + a^*T^2 + b^*T^5 + c^*exp(-T_0/T)$). The inset shows $\rho(T)$ data of crystal with *RRR* = 24,850

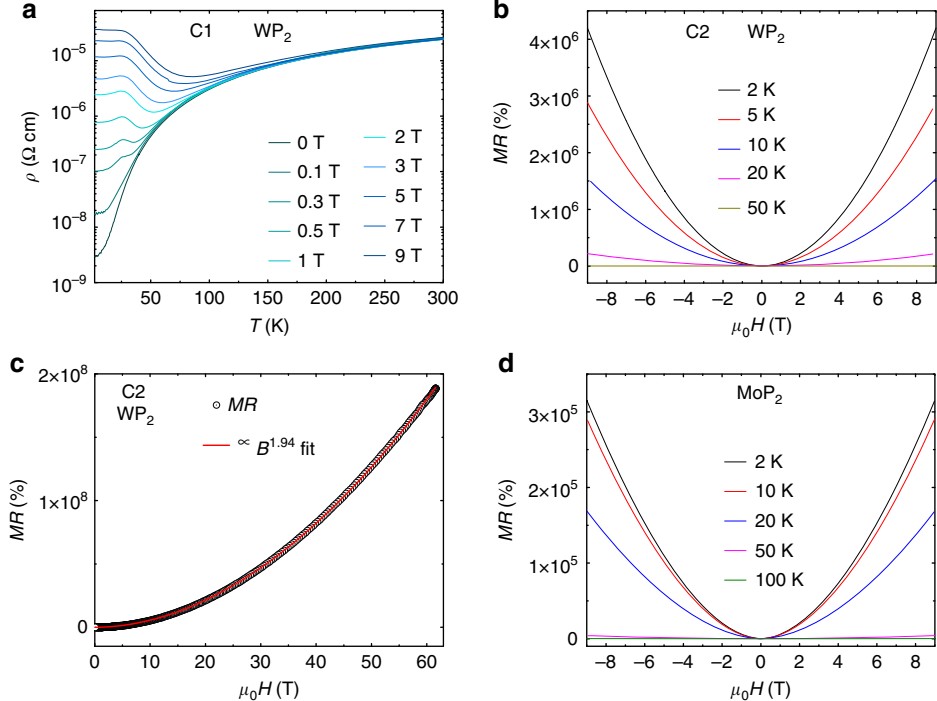

**Fig. 2** Magnetoresistance of $WP_2$ and $MoP_2$ up to 9 T in static magnetic field and up to 63 T in pulsed magnetic field. **a** $\rho(T)$ data of $WP_2$ at various magnetic fields in a temperature range of 2–300 K. **b** $\rho(B)$ data of $WP_2$ at 2 K and up to 50 K. The highest $MR$ of $4.2 \times 10^6$% is observed at 2 K and 9 T. **c** $\rho$ (B) data in a pulsed magnetic field up to 63 T and 2.5 K. The red line shows a near-perfect parabolic fit of the data up to the highest magnetic field. Extremely large $MR$ of $\sim 2 \times 10^8$% is observed. **d** $\rho(B)$ data of $MoP_2$ at 2 K and up to 100 K. The highest $MR$ of $3.2 \times 10^5$% is observed at 2 K and 9 T

**Evolution of Weyl points**. In Fig. 1c we depict how a four-fold degenerate Dirac point can be split into two Weyl points with opposite chirality via inversion symmetry breaking (or alternatively in a magnetic field). The protection of these Weyl points against annihilation depends on how far they are separated from each other in momentum space. From ab-initio calculations, in the absence of spin orbit coupling (SOC) $WP_2$ and $MoP_2$ possess two pairs of four-fold degenerate linear band crossing points with opposite sign of Chern numbers ($\pm 2$). The Chern number is the integral of the Berry curvature around a particular point in momentum space. When SOC is introduced, the spin degeneracy is lifted so that the linear band crossing points evolve into two × two-fold degenerate linear band crossing Weyl points of the same chirality. The fact that they have the same chirality means that they are robust (see Fig. 1d). This makes $WP_2$ and $MoP_2$ unique which differ from other Weyl semimetals and the effect of such robustness of Weyl points can be expected in the electrical transport. We have studied the electrical properties of $WP_2$ and $MoP_2$ together with theoretical calculations. We focus on the electronic properties of $WP_2$ and the data of $MoP_2$ are mostly included in the Supplementary Information.

**Zero field resistivity behavior**. The zero field resistivity of $WP_2$ shows a linear temperature dependence at high temperature that is indicative of dominant electron-phonon scattering (see Fig. 1e inset). For the several crystals studied (namely, C1–C5, Supplementary Fig. 4), the smallest resistivity observed at 2 K was $\rho \sim$ 3 nΩ cm yielding extremely large residual resistivity ratios (RRRs) = $\rho$ (300 K)/$\rho$ (2 K) with up to RRR ≈ 25,000. To the best of our knowledge, these very high values of the low temperature conductivity and RRR are the largest yet reported in any binary compound. $MoP_2$ also shows the similar temperature dependence of resistivity and this reaches from 25.78 μΩ cm at 300 K to

10 nΩ cm at 2 K (see Supplementary Fig. 7a) with RRR = 2578. While this may be an indication of a very high purity of the $WP_2$ crystals studied here, another more intriguing possibility is that the unique electronic properties of these compounds make certain scattering mechanisms less likely through topological protection. In order to elucidate this further, we present a detailed temperature dependence of the resistivity at low temperatures in Fig. 1e for $WP_2$. The dependence cannot be accounted for by the usual electron-electron (e-e, $T^2$-behavior) and electron-phonon scattering (e-ph, $T^5$-behavior) mechanisms. We observe that the resistivity falls more steeply as the temperature is reduced. One mechanism could be phonon drag, however, phonon drag is usually difficult to observe because it is obscured by electron-defect scattering processes[24,25]. Phonon drag gives an exponential dependence of $\rho$ (T) which we find considerably improves the fit quality to our data (blue solid line in Fig. 2a). We successfully employ this fitting scheme to other $WP_2$ crystals as well (Supplementary Fig. 5). Resistivity of $MoP_2$ at 2 K is found to be around one order of magnitude larger than $WP_2$.

**Effect of magnetic field on resistivity**. Magnetic field is a potent tool to study the motion of electrons inside metals. In general, when the electric and magnetic fields are applied transverse to each other, a positive $MR$ is observed due to the Lorentz force. In $WP_2$ despite a very high conductivity, we observe a huge magnetoresistance, as shown in Fig. 2a, which depicts $\rho$ (T) for various magnetic fields applied along the b-axis, while the current (I) was applied along the a-axis. The field dependence of the resistivity is quite small for temperatures above 100 K, below which it starts to increase drastically. For fields above 0.5 T we find that $\rho$ (T) displays an upturn below $\sim$ 50 K which is typical of many semimetals. At 9 T, this amounts to a band gap of 22 meV by fitting the low temperature $\rho$ (T) data by Arrhenius equation. In

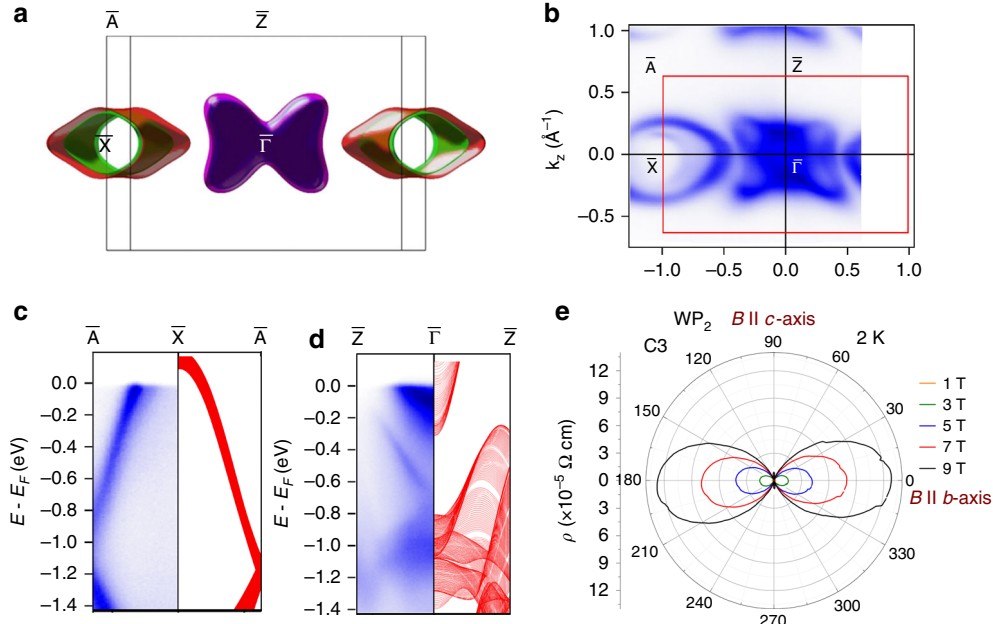

**Fig. 3** Electronic structure, ARPES and corresponding anisotropic *MR* in WP$_2$. **a** Projection of calculated Fermi surface on the *ac*-plane. Spaghetti-like open hole Fermi surfaces located around X-point in the BZ, extending along the *b*-axis. Bow-tie-like closed electron Fermi surfaces located around Y-point in the BZ. **b** Fermi surface cross section of WP$_2$ along *b*-axis from ARPES measurements showing good correspondence with the calculated Fermi surface. **c**, **d** Comparison of energy dispersions from calculation (right) and ARPES measurement (left) along $\overline{A} - \overline{X} - \overline{A}$ and $\overline{Z} - \overline{\Gamma} - \overline{Z}$, respectively. The calculated energy dispersions are projected to $k_y$. **e** Anisotropy in the resistivity due to the Fermi surface topology. *MR* is the maximum and minimum when *B* is parallel to the *b*- and *c*-axis, respectively. *I* is applied along the *a*-axis. A small misalignment of the crystal in the *bc*-plane was corrected (see Supplementary Fig. 11)

bismuth and graphite this behavior was argued to be a magnetic field-induced excitonic-insulator transition which however requires that the system is near quantum limit[26–28]. Conventional multi-band approach has also been undertaken alternatively to explain this transition in compensated semimetals[29]. The upturn is followed by a peak at lower temperatures which was also seen in bismuth and graphite. It is believed to arise from super-conducting correlations when *B* is larger than the quantum limit[27]. However, in WP$_2$ the peak appears much below the quantum limit which we estimate to be of several thousand tesla. Similar behavior of peak and upturn in $\rho$ (*T*) data was observed in many other crystals measured (see Supplementary Fig. 6). Further work is required to understand the origin of these effects.

At 2 K and 9 T, WP$_2$ exhibits a transverse *MR* of $4.2 \times 10^6$% (Fig. 2b), which is the largest yet reported in any compound and this retains up to 63 T with a value of $2 \times 10^8$%. We find that the value of *MR* decreases sharply with decreasing *RRR* values (Supplementary Fig. 8). An order of magnitude decrease in *RRR* results in a two-order of magnitude decrease in the *MR* value at 2 K and 9 T. Our measured *MR* is well described by a near parabolic field dependence, $MR \propto B^{1.94}$, up to the maximum field (63 T) explored as shown in Fig. 2c. Such an accurate scaling of *MR* with *B* makes WP$_2$ an ideally suited material for accurate magnetic field sensors (only 0.2% error due to quantum oscillations, see Supplementary Fig. 9 and Supplementary Note 2) which can be used in the megagauss regime. Moreover, MoP$_2$ also exhibits extremely large parabolic *MR* with a value of $3.2 \times 10^5$% at 2 K and 9 T (Fig. 2d). The largest MR among several measured crystals of MoP$_2$ was $6.5 \times 10^5$% (Supplementary Fig. 7) which is slightly less as compared to WP$_2$.

**Fermi surface topology**. To understand the remarkable proper-ties of WP$_2$ and MoP$_2$ further, we have performed electronic band

structure calculations based on the density functional theory. The lack of inversion symmetry in WP$_2$ leads to a spin-splitting of the bands, and both the electron and hole Fermi surfaces (FSs) come in pairs with Rashba-like splitting, see Supplementary Fig. 1. The hole and electron FSs are located around the X and Y points of the BZ, respectively, as shown in Fig. 3a. The pair of hole FSs are open and spaghetti-like extending along the *b*-axis while electrons form a pair of bow-tie-like closed FSs for WP$_2$ and MoP$_2$. From the slope of Hall resistivity vs. *B* at high magnetic field we obtain dominating hole-type carrier concentration and mobility of $5 \times 10^{20}$ cm$^{-3}$ and $4 \times 10^6$ cm$^2$ V$^{-1}$ s$^{-1}$, respectively at 2 K, while similar order of carrier density $1.2 \times 10^{21}$ cm$^{-3}$ has also been found at charge neutral point in calculation. Hall resistivity at different temperatures and the corresponding calculated carrier density and mobility of WP$_2$ are shown in Supplementary Fig. 10.

**Angle-resolved photoemission spectroscopy (ARPES)**. In order to directly investigate the electronic structure of WP$_2$, we have performed ARPES measurements on the (010) surface with photon energy $h\nu = 50$ eV. Both ARPES and theoretical results indicate that no unclosed Fermi arc exists on the (010) surface, since the projections of a pair of Weyl nodes with opposite chirality overlap with each other on the (010) surface. However, we find that the measured Fermi surface and energy dispersion match very closely to the bulk electronic band, verifying the accuracy of our band structure calculations. From AREPES measurements we can see that the FSs in (010) direction contain two types of Fermi surfaces locating around $\overline{X}$ and $\overline{\Gamma}$ point, respectively, which fit the calculated Femi surface very well (compare Fig. 3a, b). Due to the tube-shape of the hole FSs, their 2D projection along *b*-direction behaves as a closed loop without any states near the center, as seen around the X point in Fig. 3b. Our calculated electronic band structures are further checked by

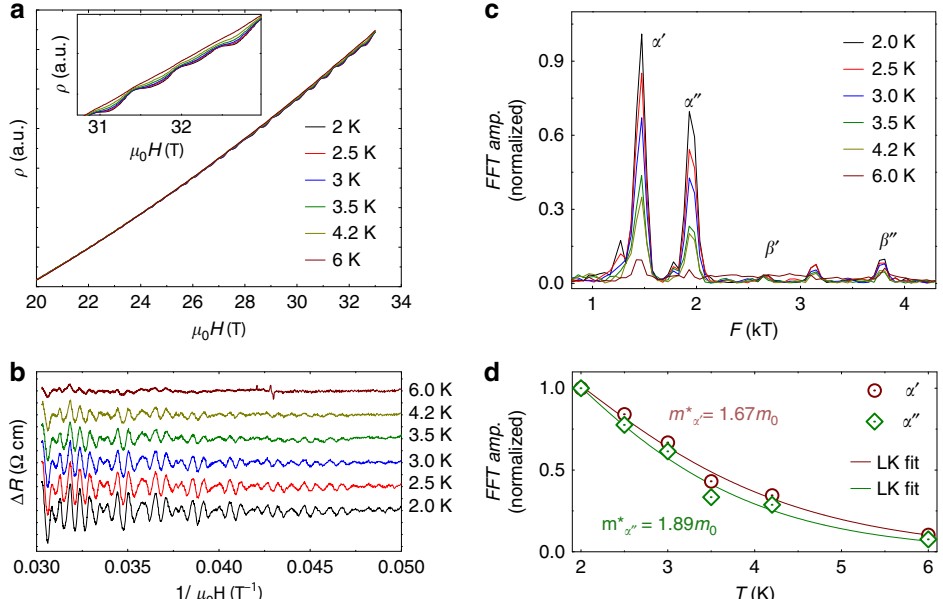

**Fig. 4** SdH oscillations in WP₂ in magnetic fields up to 33 T of C3. **a** $\rho(B)$ data at different temperatures from 2 to 6 K shows quantum oscillations. Oscillations are clearly visible in the inset with a zoomed in view at high B. **b** corresponding SdH oscillations amplitudes obtained by subtracting a continuous polynomial. **c** FFT amplitudes as a function of the temperature showing the peaks corresponding to holes and electron pockets as predicted by calculations. **d** Effective mass calculations of the hole pockets from the LK formula fits to the FFT-amplitude vs T data

the comparison with energy dispersions from AREPES in $\overline{A} - \overline{X}$ and $\overline{Z} - \overline{\Gamma}$ directions, which cross the hole and electron pockets, respectively. From Fig. 3c, we see that the calculated energy dispersion for the valence bands fit the ARPES measurements very well in the $\overline{A} - \overline{X}$ direction in a large energy window of −1.4–0 eV. The energy dispersion in the $\overline{Z} - \overline{\Gamma}$ direction contains both valence and conductions bands, as shown in Fig. 3d. Because of the small photon energy involved in the ARPES measurements, some bulk states are not observed, but for all the measured states we can find the correspondence from the calculations. Further studies on other surfaces, especially on the (001) surface, are needed to identify the possible arc states.

**Anisotropy in transport**. The angular dependence of MR of a compound is a direct reflection of the FS topology. A magnetic field, $B\|$ b-axis will lock the charge carriers with a cyclotron motion around the FSs perpendicular to b-axis and a large MR is expected. While tuning the direction of magnetic field from b- to c-axis, the perpendicular cross- section area of the FS changes smoothly and becomes infinite when the field is parallel to c-axis owing to the shape of spaghetti-type open FSs, and would lead to a dramatic drop of MR, which is consistent with the measured anisotropic MR (Fig. 3e). By contrast, the bow-tie-like electron FSs are closed pockets with smaller anisotropies perpendicular to a-axis. Thus, the anisotropic MR is mainly due to the hole FSs. Moreover, the shape of the FSs are robust over a large energy range from −0.1 to 0.1 eV, which would lead to an insensitivity of the large MR to doping. The anisotropy of the MR shown in Fig. 3b was measured by rotating the WP₂ crystal around the a-axis with a magnetic of 9 T varied within the bc-plane. The current was applied along the a-axis. The MR is maximum when the field is along the b-axis (0°) and decreases by 2.5 orders of magnitude when the field is oriented along the c-axis (90°) (Fig. 3b). Such a large anisotropy in MR is rare in a 3D compound and typically seen in 2D van der Waals compounds. Surprisingly, the effect is much more pronounced compared to 2D WTe₂

(Supplementary Fig. 11). Large anisotropy in MR is also observed in MoP₂ (Supplementary Fig. 12).

**Quantum oscillations**. The extremal cross section area of the Fermi surface perpendicular to the applied magnetic field is directly related to the frequency of the quantum oscillations. To map the FS experimentally, we have employed resistivity measurements of WP₂ in static magnetic field of 33 T along b-axis and pulsed fields of 65 T along b-axis for MoP₂. Figure 4a shows the resistivity of WP₂ at different temperatures between 2 and 6 K. The Shubnikov–de Haas oscillations (SdH) are clearly visible by subtracting a cubic polynomial from the resistivity data. The extracted amplitudes of the SdH oscillations in WP₂ as a function of the inverse magnetic field are shown in Fig. 4b. The fast Fourier transform (FFT) of the SdH oscillations identifies four fundamental frequencies which are identified from the spaghetti-type holes FS ($\alpha'$ at 1460 T and $\alpha''$ at 1950 T) and from the bow-tie-type electrons FS ($\beta'$ at 2650 T and $\beta''$ at 3790 T) with the help of ab-initio calculations. The quantum oscillations were calculated from the k-space areas of the extremal cross-sections of the FSs with a magnetic field along y. We found four frequencies at ~ 1300 T, 1900 T from 2 hole like FSs, and 2800 and 3900 T from 2 electron like FSs, which fit the experimental results well, as presented in Fig. 4c. We calculate the effective mass ($m^{\star}$) of the electrons in $\alpha'$ and $\alpha''$ pockets from the temperature dependence of the SdH amplitudes (Fig. 4d) using the Lifshitz–Kosevich (LK) formula: $\Delta R = X/\sinh(X)$, where $X = 14.69\, m^{\star}T/B$ and $B$ is the average field. $m^{\star}$ for holes in $\alpha'$ and $\alpha''$ pockets are $1.67 m_0$ and $1.89 m_0$, respectively. For the electrons in $\beta'$ pocket, $m^{\star}$ is $1.32 m_0$ (see Supplementary Fig. 13), however we could not obtain $m^{\star}$ of $\beta''$ because of the small amplitude associated with it. If we consider circular Fermi surface cross section of $\alpha'$- hole band along the b-axis which is a fair approximation to make, we can calculate the Fermi area $A_F$ from the Onsager relation: $F = (\hbar/2\pi e)A_F$ to be 0.14 Å⁻². This gives rise to the Fermi vector $k_F$ of 0.21 Å⁻¹ and matches quite well to the Fermi cross section of this band normal

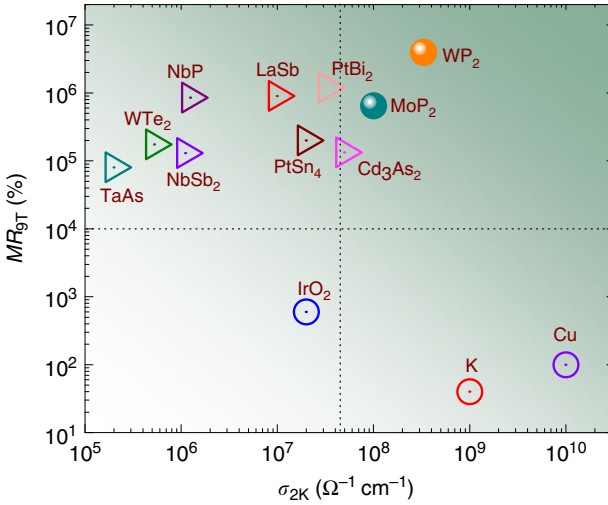

**Fig. 5** A comparison of *MR* and conductivity of some well-known metals and semimetals WP₂ and MoP₂ are placed in the *MR*-conductivity plane at 2 K and 9 T along with some well-known metals and semimetals for comparison. Semimetals are denoted by triangles, metals by hollow circles and WP₂ and MoP₂ by solid circles. Metals with high conductivity have smaller *MR* and semimetals with smaller conductivities have larger *MR*. WP₂ and MoP₂ exhibit both very large conductivity, as well as extremely high *MR*

to the *b*-axis as observed in the ARPES measurements of WP₂. A very large Fermi velocity of $1.4 \times 10^5$ m/s is obtained from the relation, $v_F = \frac{\hbar k_F}{m^*}$. Similarly, for MoP₂, we employ pulsed magnetic field up to 63 T to study the SdH oscillations which agree well with the calculated Femi surfaces (Supplementary Fig. 14).

## Discussion

We now consider the origin of large conductivity and *MR* in WP₂. The Weyl point induced spin texture (Berry phase) (Supplementary Note 1 and Supplementary Figs. 1 and 2) can effectively suppress the backscattering. The robustness of these Weyl points in WP₂ due to same chirality of the neighboring Weyl nodes will enhance such effect. The experimental proof of the suppression comes from the ratio, $r$ of transport lifetime ($\tau_{tr}$) and quantum lifetime ($\tau_q$) of scattering. $\tau_{tr}$ is calculated from the Drude model as $\tau_{tr} = \mu m^*/e = 3.8 \times 10^{-9}$ s, where $\mu$ is the mobility and $m^*$ is the effective mass of $\alpha'$ band. This also gives rise an extraordinarily large classical mean free path of 0.5 mm. $\tau_q$ is obtained from the broadening of the SdH oscillations as $\tau_q = \hbar/(2\pi T_D)$, where $T_D$ is the Dingle temperature. With $T_D$ of 1.54 K for $\alpha$ band we obtain $\tau_q = 7.9 \times 10^{-13}$ s. The ratio $r = 5000$ thus indicates the large suppression of the backscattering of carriers which is comparable to Cd₃As₂[1]. The large value of $r$ also indicates the fact that the momentum conserving processes (electron–electron scattering and phonon drag) are in balance with the momentum relaxing processes (electron-defect, electron phonon, Umklapp scatterings) making WP₂ a good candidate for observing hydrodynamic flow of electrons. In fact, we have observed a clear signature of hydrodynamic flow in WP₂ by undertaking size dependent transport measurements[30]. Therefore, hydrodynamic effects can play significant role in the large conductivity in WP₂. Moreover, we cannot also rule out the effect of SOC induced spin splitting. Interestingly, MoP₂ with smaller SOC, exhibits one order of less conductivity compared to WP₂. The large conductivity in WP₂ at low temperature ensures big *RRR* value. The unusually large value of *RRR* has a significant role

towards the enhancement of *MR*. Recently, in Dirac semimetal PtBi₂, it was shown that the large *RRR* value is one of the main factors for large *MR*[31]. Carrier compensation in semimetals also gives rise to large parabolic *MR*. Our first principles calculations predict equisized electron and hole pockets, which was also confirmed by Fermi surface obtained from the ARPES measurements. In order to further verify, we fit our low temperature Hall conductivity to two band model (see Supplementary Fig. 10 for details). This provides a near compensation of holes ($1.5 \times 10^{20}$ cm⁻³) and electrons ($1.4 \times 10^{20}$ cm⁻³) at 2 K. Hence, the large mobility, extremely large *RRR*, charge compensation all contribute to the ultra-high nonsaturating parabolic *MR* in WP₂.

Having seen the extremely large *MR* and conductivity in WP₂ and MoP₂, we compare these quantities with several topological metals and other well-known and highly conducting trivial metals in Fig. 5. WP₂ and MoP₂ perform much better than Dirac semimetal Cd₃As₂[1] which also exhibits large *MR* and conductivity. Other semimetals like NbP, WTe₂, TaAs, and so on[2,32] where the *MR* is quite large, conductivity is orders of magnitudes smaller because of small carrier concentrations. In copper, which is one of the most conductive metals known, the *MR* is small and is of the order of 50–250 % in single crystals with *RRR* = 40,000–62,000[33]. Another class of highly conducting materials with very large *RRR* are the rutile and delafossite oxides such as IrO₂[34] and PdCoO₂[35]. Here the *MR* is very low compared to semimetals, which typically, however, have low conductivities due to their small carrier concentrations, for example, NbP[2] and NbSb₂[36]. In conclusion, WP₂ and MoP₂ have conductivities comparable to those in metals like copper while still exhibiting *MR* values more than any Dirac or Weyl semimetals known.

Although, chiral pumping of charge between the Weyl nodes of opposite chiralities (chiral anomaly) are possible in WSM-IIs, it is much more difficult to detect this in WSM-IIs compared to standard Weyl semimetals because it can only be observed when parallel electric and magnetic fields are applied along certain crystal directions. We do not observe any negative MR when we apply B and I along *a*-axis in WP₂ (see Supplementary Fig. 15). WP₂ is predicted to exhibit the effect of chiral anomaly only when both electric and magnetic fields are applied along *b*-axis[23]. The as-grown crystals of WP₂ are all needle-shaped with their length aligned along *a*-axis, which, therefore, makes it very difficult to apply the electric field along *b*-axis. Another difficulty to observe the chiral anomaly in WP₂ is its extremely large positive *MR* when the field is aligned along *b*-axis. A slight disorientation of the field from the *b*-axis results in a large positive *MR* which would make the observation of the chiral anomaly even more difficult. We believe that these limitations can be overcome by using a focussed ion beam to fabricate a better sample.

In conclusion, we have shown that WP₂ is a remarkable compound with properties unlike any other compound yet studied in the families of Dirac, Weyl and novel fermion materials. It displays record breaking *RRR* values and ultra-high low temperature conductivities and a non-saturating magnetoresistance. One of the most interesting questions is the degree to which the topological electronic properties of this material account for its unusual properties. We observe a large suppression of backscattering of electrons and, considering the fact that no special procedures were used to purify the elemental starting materials, we conjecture that the protection of the Weyl points from annihilation plays an important role. This will be an important focus of future work.

## Methods

**Single crystals growth**. Crystals of WP₂ were prepared by chemical vapor transport method. The single crystals of WP₂ and MoP₂ were grown by chemical vapor transport. Starting materials were red phosphorous (Alfa-Aesar, 99.999%)

and tungsten/molybdenum trioxide (Alfa-Aesar, 99.998%) with iodine as a transport agent. The materials were taken in an evacuated fused silica ampoule. The transport reaction was carried out in a two-zone-furnace with a temperature gradient of 1000 °C (T1) to 900 °C (T2) for serval weeks[37]. After reaction, the ampoule was removed from the furnace and quenched in water. The metallic-needle crystals were characterized by X-ray diffraction (see Supplementary Fig. 3).

**Electrical transport measurements**. Resistivity measurements were performed in a physical property measurement system (PPMS-9T, Quantum Design) using the ACT and Resistivity with rotator option. For longitudinal resistivity, linear contacts were made on the naturally grown crystals by silver paint and 25 μm platinum wires. The longitudinal and Hall resistivity were measured in 4-wires and 5-wires geometry, respectively using a current of 3.0–5.0 mA at temperature range from 2 to 300 K and magnetic fields up to 9 T.

4-point resistivity measurements at high static magnetic field were performed at HFML, Nijmegen, Netherlands. The sample was placed on a commercially supplied chip carrier (insulated using a layer of cigarette paper). 25 μm gold wire and 4929 silver paste were used to make contacts between the chip carrier and the sample. An AC current of 1 mA (using a Keithley 6221 current source) was applied along the *a*-axis, and the voltage was measured along the same direction using a Stanford Research SR 830 lock-in amplifier at a frequency of 13 Hz. The sample temperature was controlled by a 4He flow-cryostat, and applied fields up to 33 T were generated using a resistive Bitter magnet available at the HFML. The high pulsed field-dependent resistivity was measured in a four point geometry using a 62 T non-destructive pulsed magnet driven by a capacitor bank at the Dresden High Magnetic Field Laboratory. The excitation current was 1 mA with a frequency of 3333 and 7407 kHz.

**ARPES measurements**. ARPES measurements were performed with VG-Scienta R4000 electron analyzers at SIS beamline at Swiss Light Source, Paul Scherrer Institut. The energy and angular resolutions were set at 15 meV and 0.2°, respectively. Samples were cleaved in-situ along the (010) crystal plane in an ultrahigh vacuum of $5 \times 10^{-11}$ Torr. A shiny mirror-like surface was obtained after cleaving the samples, confirming their high quality. The Fermi level of the samples was referenced to that of a gold film evaporated onto the sample holder.

**Band structure calculations**. The electronic structures were calculated by the ab-initio calculations based on the density functional theory. We have used the projected augmented wave method as implemented in the program of Vienna ab-initio Simulation Package (VASP)[38]. For getting accurate band structures the exchange and correlation energy was considered in the modified Becke-Johnson (MBJ) exchange potential[39,40]. Fermi surfaces were interpolated in a dense *k*-grids of $500 \times 500 \times 500$ points by using maximally localized Wannier functions[40].

**Data availability**. The data that support the findings of this study are available from the corresponding authors N.K. and C.F. upon request.

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

## Acknowledgements

This work was financially supported by the ERC Advanced Grant No. (742068) "TOP-MAT". We acknowledge Prof. Stuart Parkin for fruitful discussions. We acknowledge the support of the High Field Magnet Laboratory Nijmegen (HFML-RU/FOM), and High Magnetic Field Laboratory Dresden (HLD) at HZDR members of the European Magnetic Field Laboratory (EMFL). The ARPES studies in the work was supported by NCCR-MARVEL funded by the Swiss National Science Foundation.

## Author contributions

N.K., C.S., and C.F. designed the experiments. N.K. and C.S. performed transport measurements. V.S. and M.S. grew single crystals. K.M. and H.B. performed Laue x-ray diffraction experiments. I.L., O.Y., U.Z., and T.F. performed high magnetic field transport measurements. Y.S. with inputs from B.Y. carried out theoretical calculations. N.X., M.Y.,

and M.S. performed ARPES measurements. N.K., C.S., and C.F. wrote the manuscript with inputs from all the authors.

## Additional information

**Competing interests:** The authors declare no competing financial interests.

