## [Peer Review File · Nature Communications]

Reviewers' comments:

Reviewer #1 (Remarks to the Author):

The manuscript reported the extreme magnetoresistance (XMR) and angle resolved photoemission spectroscopy measurements on both semimetallic WP₂ and MoP₂ crystals. The WP₂ crystals show high quality with an extremely large residual resistance ratio ~25000, and all of the transport data are very similar to those observed in other Dirac or WSMs. The most crucial result is the observation of extremely large highly anisotropic magnetoresistance above 200 million % at 63 T and 2.5 K. As claimed by the authors that this MR value in WP₂ is the biggest one compared with those reported in literature for any other compounds. The data is high quality and well done.

However, the paper did not make any convincing discussion on the mechanism of such huge XMR. What are the key factors for this XMR? such as the high mobility or high carriers concentration or the complete electron and hole compensation? From the band calculation and the experimental data, both the hole or electron carriers mass are much larger than a free electron mass, the Weyl points are far below the Fermi level. I don't think such huge XMR is related to any Weyl fermions. This is because the transport properties are quite similar to other semimetals showing similar XMR. Strictly speaking, the nature of the XRM in WP₂ or WMo₂ perhaps is the same as in other semimetals. Maybe the XMR here is due to the sample high quality, where the defect scattering is less important and thus result in huge electron mean free path.

In summary, the manuscript did not show insight on understanding the origin of the XMR in Weyl semimetals although it shows the largest XMR properties in WP₂. At least, to my point of view, it is insufficient for supporting publication in Nature Communication by only reporting the huge XMR data. In fact, the reported MR value is also on the order of those reported in other Weyl semimetals.

Reviewer #2 (Remarks to the Author):

The authors present experimental measurements showing an extremely high magnetoresistance and a large suppression of backscattering of electrons in the type-II Weyl semimetals WP₂ and MoP₂. While I agree with the interesting data, I am circumspect regarding the explanation. Overall, the paper is basically clear and their results are

presented well. However, there are several points that I would recommend the authors address in order to make their paper clearer:

- 1) It would be clearer if the authors can explain why the small fluctuation in the MR curves of MoP₂ in Fig. 2d.
- 2) In the paper the authors state that the upturn is followed by a peak at lower temperatures, without explanation. In my opinion, this explanation needs to be given in more detail for it to be meaningful.
- 3) The MR in WP₂ exhibits a near quadratic field dependence while a parabolic MR in MoP₂, are they similar?
- 4) In Ref. 20 they mentioned that the predicted Fermi arcs span extended regions of k space and are localized below the Fermi level. They should be readily observable in ARPES experiments on a cleaved (001) surface of XP2 compounds. The author should replenish ARPES measurements on the (001) surface.
- 5) The authors say that the SOC in MoP₂ is one order smaller compared to the WP₂, is there any experimental evidence?

Reviewer #3 (Remarks to the Author):

In this manuscript, the authors studied the electronic properties of type-II Weyl semimetals WP₂ and MoP₂. They observed the extremely low residual resistivity and highly anisotropic magnetoresistance in high quality WP₂ samples. They attributed the former to the large suppression of charge carrier backscattering due to the robust Weyl points, and the latter to the anisotropic Fermi surface which was also investigated by ARPES measurements and first principle calculations in this manuscript. The observed results on extremely high magnetoresistance and conductivity in WP₂ with robust Weyl points are exciting. I would recommend its publication on Nature Communications after the authors address my following comments:

In the manuscript, the author observed the extremely large MR in WP₂, which also decreased sharply with decreasing RRR values (Fig. S8). However, the intrinsic physics of the formation and change of the large MR have been barely discussed. For another type-II Weyl semimetal material WTe₂ with similar band structures, charge compensation and high mobility play important roles in the large non-saturating MR. Do these factors influence the observed MR in WP₂? Moreover, it is imprecise to calculate the carrier density and mobility by single-band

model due to the nonlinear hall resistivity. Two-band model should be necessary for the calculations. In addition, does the suppression of backscattering have correlation with the observed large MR? It is necessary for the authors to discuss the formation of large MR in details in the revised manuscript.

And also, in the sentence ‘The archetypical WSM-IIs are the two dimensional van der Waals compounds WTe₂ and MoTe₂ [14][15][16]’, the cited references only include the theoretical work on MoTe₂ [14] and the ARPES works on WTe₂ [15] and MoTe₂ [16]. Few other important works from the community should be properly cited as well, such as the theoretical work on WTe₂ [Nature 527, 495-498 (2015)] and the transport work of chiral anomaly in WTe₂ [Nat. Commun. 7,13142 (2016)].

Response to referees

Reviewer #1 (Remarks to the Author):

The manuscript reported the extreme magnetoresistance (XMR) and angle resolved photoemission spectroscopy measurements on both semimetallic WP₂ and MoP₂ crystals. The WP₂ crystals show high quality with an extremely large residual resistance ratio ~ 25000 , and all of the transport data are very similar to those observed in other Dirac or WSMs. The most crucial result is the observation of extremely large highly anisotropic magnetoresistance above 200 million % at 63 T and 2.5 K. As claimed by the authors that this MR value in WP₂ is the biggest one compared with those reported in literature for any other compounds. The data is high quality and well done.

We thank the reviewer for recognizing our data of high quality.

However, the paper did not make any convincing discussion on the mechanism of such huge XMR. What are the key factors for this XMR? such as the high mobility or high carriers concentration or the complete electron and hole compensation?

We agree with the reviewer that we have not expanded on the origin of large MR in WP₂ and MoP₂, as we focused more on the excellent conductivity which in turn is related to the large MR in our compounds.

As already discussed in the text, the extremely large conductivity has possible contributions from the high mobility, spin texturing because of robust topology, hydrodynamic flow of electrons at low temperature etc. The unusually large value of RRR in WP₂ has a significant role towards enhancement of MR. First principle calculations predict equi-sized electron and hole pockets, which was also confirmed by Fermi surface obtained from the ARPES measurements. In order to further verify, we fit our low temperature Hall conductivity to two band model (see SI for details). This provides a near compensation of holes ($1.5 \times 10^{20} \text{ cm}^{-3}$) and electrons ($1.4 \times 10^{20} \text{ cm}^{-3}$) at 2 K. Hence, the large mobility, extremely large RRR, charge compensation all contribute to the ultra-high MR in WP₂.

We have now added a paragraph in the revised manuscript explaining the origin of large MR in WP₂.

From the band calculation and the experimental data, both the hole or electron carriers mass are much larger than a free electron mass, the Weyl points are far below the Fermi level. I don't think such huge XMR is related to any Weyl fermions. This is because the transport properties are quite similar to other semimetals showing similar XMR. Strictly speaking, the nature of the XMR in WP₂ or WMo₂ perhaps is the same as in other semimetals. Maybe the XMR here is due

to the sample high quality, where the defect scattering is less important and thus result in huge electron mean free path.

We agree with the reviewer that the carriers' mass are more than the free electron mass and the Weyl points are situated further below the Fermi level. We propose that the large conductivity in the material originate from several factors and the occurrence of Weyl points being one of them. Another important factor is the hydrodynamic flow of electrons in these systems which we have proved in arXiv: 1706.05925. The large spin orbit coupling induced spin splitting which is also confirmed from the quantum oscillations and ARPES data, coupled with the spin texturing provides a huge suppression of backscattering of carriers. All these factors paly significant role in the large conductivity. The large RRR (i.e. extremely small ρ_0) thus obtained is crucial for the observation of large ratio $(\rho_B/\rho_0) \sim MR$.

In summary, the manuscript did not show insight on understanding the origin of the XMR in Weyl semimetals although it shows the largest XMR properties in WP₂. At least, to my point of view, it is insufficient for supporting publication in Nature Communication by only reporting the huge XMR data. In fact, the reported MR value is also on the order of those reported in other Weyl semimetals.

We disagree with the reviewer here. WP₂ in addition to XMR, also exhibits following unique properties:

- 1) Extremely large conductivity comparable to pure copper single crystals.
- 2) Extremely long sub-millimeter mean free path comparable to bismuth.
- 3) Extremely large anisotropic *MR* of 2-3 order magnitude, unique to any Weyl or Dirac semimetal
- 4) Extremely large ratio of transport and quantum mean free path (5000), indicating a strong suppression of backscattering.
- 5) Near-parabolic scaling of the MR until 65 T makes it ideal candidate for a megagauss sensor.

We hope the reviewer finds the revised version of the manuscript acceptable for Nature communications.

Reviewer #2 (Remarks to the Author):

The authors present experimental measurements showing an extremely high magnetoresistance and a large suppression of backscattering of electrons in the type-II Weyl semimetals WP₂ and MoP₂. While I agree with the interesting data, I am circumspect regarding the explanation. Overall, the paper is basically clear and their results are presented well.

However, there are several points that I would recommend the authors address in order to make their paper clearer:

We thank the referee for commending our data and their presentation.

1) It would be clearer if the authors can explain why the small fluctuation in the MR curves of MoP₂ in Fig. 2d.

We thank the referee for pointing out the small fluctuations in the data of MoP₂. These occurred due to the current induced small temperature fluctuations. We now replace the old data with the more smooth data with negligible temperature-induced fluctuations.

2) In the paper the authors state that the upturn is followed by a peak at lower temperatures, without explanation. In my opinion, this explanation needs to be given in more detail for it to be meaningful.

We thank the referee for the suggestion. We now include a discussion about the low temperature resistivity upturn and the peak in the revised manuscript.

3) The MR in WP₂ exhibits a near quadratic field dependence while a parabolic MR in MoP₂, are they similar?

We thank the referee for pointing out the discrepancy. Both WP₂ and MoP₂ show near parabolic behavior. We have fitted the MR curve by: $MR = a + b * B^c$, without using any linear term.

4) In Ref. 20 they mentioned that the predicted Fermi arcs span extended regions of k space and are localized below the Fermi level. They should be readily observable in ARPES experiments on a cleaved (001) surface of XP₂ compounds. The author should replenish ARPES measurements on the (001) surface.

We agree with the reviewer that the Fermi-arc should be visible on the (001) surface. We tried to cleave many crystals (20-30) but the crystals naturally cleave along (010). Hence, we could not yet observe the Fermi arc states in WP₂.

5) The authors say that the SOC in MoP₂ is one order smaller compared to the WP₂, is there any experimental evidence?

From the sentence "*the conductivity of MoP₂ with relatively small SOC is one order smaller compared to WP₂*" we meant that MoP₂ with smaller spin-orbit coupling has one order of less conductivity compared to WP₂. We have modified the sentence to avoid any confusion.

However, the experimental evidence of the larger SOC in WP₂ compared to MoP₂ comes from the quantum oscillations analysis. We find the larger separation between two hole pockets frequencies in WP₂ (~500 T) compared to MoP₂ (~380 T) when the field is applied along *b*-axis. The more direct evidence would come from ARPES measurements. While we have data for WP₂, we do not have data for MoP₂ to compare due to extremely thin needle shaped single crystals.

Reviewer #3 (Remarks to the Author):

In this manuscript, the authors studied the electronic properties of type-II Weyl semimetals WP₂ and MoP₂. They observed the extremely low residual resistivity and highly anisotropic magnetoresistance in high quality WP₂ samples. They attributed the former to the large suppression of charge carrier backscattering due to the robust Weyl points, and the latter to the anisotropic Fermi surface which was also investigated by ARPES measurements and first principle calculations in this manuscript. The observed results on extremely high magnetoresistance and conductivity in WP₂ with robust Weyl points are exciting. I would recommend its publication on *Nature Communications* after the authors address my following comments:

We thank the reviewer for recommending our work to be published in *Nature Communications*.

In the manuscript, the author observed the extremely large MR in WP₂, which also decreased sharply with decreasing RRR values (Fig. S8). However, the intrinsic physics of the formation and change of the large MR have been barely discussed. For another type-II Weyl semimetal material WTe₂ with similar band structures, charge compensation and high mobility play important roles in the large non-saturating MR. Do these factors influence the observed MR in WP₂? Moreover, it is imprecise to calculate the carrier density and mobility by single-band model due to the nonlinear Hall resistivity. Two-band model should be necessary for the calculations. In addition, does the suppression of backscattering have correlation with the observed large MR? It is necessary for the authors to discuss the formation of large MR in details in the revised manuscript.

We have now added a paragraph in the manuscript which explains the origin of large MR in WP₂. Indeed, like WTe₂, the charge compensation plays a significant role in the non saturating MR behavior. The MR remains parabolic up to 63 T field. We have now fitted the Hall conductivity to the two band model and find a near compensation of charges in WP₂. The carrier density and mobilities are comparable to that obtained from single band model.

The suppression of backscattering results in large conductivity and high RRR (i.e. extremely small ρ_0) thus obtained is crucial for the observation of large MR ($\sim \rho_B/\rho_0$).

And also, in the sentence ‘The archetypical WSM-IIIs are the two dimensional van der Waals compounds WTe₂ and MoTe₂ [14][15][16]’, the cited references only include the theoretical work on MoTe₂ [14] and the ARPES works on WTe₂ [15] and MoTe₂ [16]. Few other important works from the community should be properly cited as well, such as the theoretical work on WTe₂ [Nature 527, 495-498 (2015)] and the transport work of chiral anomaly in WTe₂ [Nat. Commun. 7,13142 (2016)].

We thank the reviewer for suggesting the relevant references for our manuscript. We have now included them in the revised manuscript.

Reviewers' Comments:

Reviewer #1 (Remarks to the Author):

The authors have made corresponding revision and answered most of my concerns. Although I do not completely satisfied with the discussion on the XMR, I have to admit that it is really difficult to give a very clear answer just based on the field dependent transport data because of the complexity of the band structures. The interpretations on the XMR in the present revised version sound reasonable. Considering the rapidly moving nature of this field, I am happy to recommend it for publication after a minor revision.

1) In the revised version, I found the authors add ref. 29 the newly published paper on PtBi₂, this materials also present XMR on the order of 10^7 at 32T. Both materials show very similar behavior with strong anisotropy and high quality. I suggest the authors should put the MR data of PtBi₂ in Fig. 5 for comparison. Maybe both materials show the same order on XMR .

2) In line 202, Pd should be Pt.

3) Between line 262-398, the “methods and DFT method” sections are in the wrong position.

Reviewer #2 (Remarks to the Author):

If compared to its initial submission, the revision has been improved and the goals are clearly stated. The authors have also addressed the points raised by the referees. As such the presentation now appears satisfactory, therefore, I recommend that this paper can be published.

Reviewer #3 (Remarks to the Author):

The authors have addressed all my questions. The revised manuscript could be now accepted by Nature Communications.

Referee 1

The authors have made corresponding revision and answered most of my concerns. Although I do not completely satisfied with the discussion on the XMR, I have to admit that it is really difficult to give a very clear answer just based on the field dependent transport data because of the complexity of the band structures. The interpretations on the XMR in the present revised version sound reasonable. Considering the rapidly moving nature of this field, I am happy to recommend it for publication after a minor revision.

Reply: We thank the referee for accepting our manuscript.

1) In the revised version, I found the authors add ref. 29 the newly published paper on PtBi₂, this materials also present XMR on the order of 10^7 at 32T. Both materials show very similar behavior with strong anisotropy and high quality. I suggest the authors should put the MR data of PtBi₂ in Fig. 5 for comparison. Maybe both materials show the same order on XMR.

Reply: We have now included the data of PtBi₂ in Fig. 5. WP₂ is still the best material in terms of MR and conductivity among all semimetals.

2) In line 202, Pd should be Pt.

Reply: We thank the referee for pointing out this typo. We have corrected it.

3) Between line 262-398, the "methods and DFT method" sections are in the wrong position.

Reply: Thank you, we have adjusted the position.

Referee 2

If compared to its initial submission, the revision has been improved and the goals are clearly stated. The authors have also addressed the points raised by the referees. As such the presentation now appears satisfactory, therefore, I recommend that this paper can be published.

Reply: We thank the referee for accepting our manuscript.

Referee 3

The authors have addressed all my questions. The revised manuscript could be now accepted by Nature Communications.

Reply: We thank the referee for accepting our manuscript.